# Embryonic Development and Survival of Siberian Sturgeon × Russian Sturgeon (*Acipenser baerii* × *Acipenser gueldenstaedtii*) Hybrids Cultured in a RAS System

**DOI:** 10.3390/ani13010042

**Published:** 2022-12-22

**Authors:** Dorota Fopp-Bayat, Tomasz Ciemniewski, Beata Irena Cejko

**Affiliations:** 1Department of Ichthyology and Aquaculture, Faculty of Animal Bioengineering, University of Warmia and Mazury in Olsztyn, 10-719 Olsztyn, Poland; 2Department of Salmonid Research, Stanisław Sakowicz Inland Fisheries Institute, 10-719 Olsztyn, Poland

**Keywords:** *Acipenser*, artificial reproduction, hybrids, embryo survival

## Abstract

**Simple Summary:**

Sturgeon (Acipenseridae) fish are a valuable fish in aquaculture not only because of the taste of their meat, but also because of the caviar obtained from the production process. Sturgeon species differ from teleosts in the main stages of the life cycle, in particular embryonic development. Therefore, research regarding the controlled reproduction and egg incubation are key during breeding and hybridization programs of sturgeon species. Moreover, sturgeon hybrids are characterized by high values of performance, breeding, and reproductive parameters, and their rearing is often more economically efficient than the rearing of purebred species Therefore in this study we analyzed selected stages of embryonic development in Siberian sturgeon (*Acipenser baerii*) and hybrids of Siberian sturgeon and Russian sturgeon (*Acipenser baerii* × *Acipenser gueldenstaedtii*). The embryo survival rate was higher in group B hybrids i.e., ♀ [Siberian sturgeon × Russian sturgeon] × ♂ Siberian sturgeon than in the control group (Siberian sturgeon), whereas group A hybrids i.e., ♀ [Siberian sturgeon × Russian sturgeon] × ♂ Siberian sturgeon were characterized by lower survival rates than the group C embryos (control). These results suggest that the survival of hybrid embryos could be determined by individual traits, such as gamete (eggs and sperm) quality and genetic factors.

**Abstract:**

The aim of this study was to describe the selected stages of embryonic development in Siberian sturgeon (*Acipenser baerii*) and hybrids of Siberian sturgeons and Russian sturgeons (*Acipenser baerii* × *Acipenser gueldenstaedtii*). For this purpose, embryos representing nine distinct developmental stages (stage 1—2.0 hpf, stage 2—5.5 hpf, stage 3—13.0 hpf, stage 4—20.0 hpf, stage 5—24.0 hpf, stage 6—26.0 hpf, stage 7—35.0 hpf, stage 8—55.0 hpf, and, stage 9—160.0 hpf; hpf—hours postfertilization) were sampled from each group (group A, group B, and group C) during incubation. Stages of embryogenesis were identified based on a 30-point scale of embryonic development in sturgeons. A total of 13 developmental stages were identified, including early cleavage, blastula formation, early and late gastrulation, onset of neurulation, beginning of organogenesis, and prelarvae. During gastrulation, the survival of hybrid embryos was highest in group B (93.8%) and lowest in group A (86.7%). Embryonic deformation was not observed during experimental incubation. The archived data relating to the embryonic development of Siberian sturgeon × Russian sturgeon hybrids could be applied to identify the individual stages of embryogenesis in hybrid sturgeons during egg incubation.

## 1. Introduction

In the northern hemisphere, sturgeons (Acipenseridae) have been long regarded as valuable fishery resources, mainly in caviar production. In recent decades, anthropogenic and industrial activities posed a serious threat to the natural habitats of wild sturgeon populations [1]. Most sturgeon species have a critically endangered status and they have been listed in the Convention on International Trade in Endangered Species of Wild Fauna and Flora (CITES). As a result, sturgeons farmed for meat and other products (such as caviar) have to be reared in aquaculture [2,3]. 

The first ichthyological research to support natural sturgeon populations involved artificial breeding and broodstock production (mostly fry) for open-water stocking. In successive stages, the scope of the research was expanded to include fry and fingerling rearing under controlled conditions, the selection of breeding candidates, and the establishment of the sturgeon spawning stock. In vivo methods for collecting gametes were developed, which enabled multiple uses of the same spawners [4,5], selective breeding, and interspecific hybridization. Hybridization among members of the family Acipenseridae is a natural phenomenon, and it occurs mainly due to a decrease in the size of spawning grounds and spawning time overlap [6]. Most intraspecific hybrids are fertile, which prompted ichthyologists to develop hybrids under controlled conditions. The main aim of these breeding efforts was to produce interspecific hybrids (displaying the heterosis effect) with higher survival and growth rates, as well as other desirable traits characteristic of the parent species. Breeding experiments confirmed that these goals had been met, and F_1_ hybrids produced by crossing species with a similar number of chromosomes proved to be particularly successful. The following F_1_ hybrids were obtained: beluga × sterlet, Siberian sturgeon × Russian sturgeon, as well as backcrossed hybrids, such as [Siberian sturgeon × Russian sturgeon] × Siberian sturgeon [7]. Russian sturgeon and Siberian sturgeon hybrids are particularly attractive due to their high meat quality and higher carcass yield in comparison with the parent species. In aquaculture, these hybrids are also characterized by a higher growth rate than the parent species, in particular the male parent [5]. Artificial breeding and embryo incubation, including embryonic development and larval hatching, are the most important stages during sturgeon rearing in aquaculture. Embryogenesis and successive developmental stages should be closely monitored to identify any irregularities. 

In Poland, various sturgeon species and hybrids are artificially bred to produce broodstock for further rearing. Therefore, a sound knowledge of embryonic development in purebred sturgeon and hybrids is needed to monitor all stages of the process and detect any abnormalities. The aim of this study was to analyze the embryonic development of backcrossed hybrids obtained by fertilizing eggs collected from Siberian sturgeon × Russian sturgeon (*Acipenser baerii* Brandt, 1869 × *Acipenser gueldensaedtii*, Brandt and Ratzeburg, 1833) hybrid females with sperm collected from Siberian sturgeon males. Embryonic development stages were compared in backcrossed hybrids and purebred Siberian sturgeons.

## 2. Materials and Methods

### 2.1. Materials

Embryos (hybrids or pure species) for the study were obtained from female A and female B—Siberian sturgeon × Russian sturgeon hybrids (hybrids), and female C—purebred Siberian sturgeon (control group). Eggs collected from all three females were fertilized with sperm collected from Siberian sturgeon males.
A and B → ♀ [Siberian sturgeon x Russian sturgeon] × ♂ Siberian sturgeon—hybrids
C → ♀ Siberian sturgeon x ♂ Siberian sturgeon—pure species

### 2.2. Controlled Breeding and Embryo Incubation

Sturgeons were bred in the Wąsosze Fish Farm, near Konin. One Siberian sturgeon female, two Siberian sturgeon x Russian sturgeon hybrid females, and three Siberian sturgeon males were prepared for breeding. The fish were initially subjected to environmental stimulation and then stimulated with hormones upon the achievement of gonadal maturity stage IV [8]. Females were injected with carp pituitary extract (5 mg kg^−^^1^), and males were treated with the Ovopel hormonal treatment at 1 granule per kg^−^^1^. Eggs were sampled in vivo, and sperm was collected by a syringe with an elastic cannula. Sperm quality was evaluated under a microscope to evaluate the sperm activated in water. Eggs were fertilized by the semidry method with sperm diluted in water at a 1:50 ratio. Fertilized eggs were incubated at 16 °C (±0.5 °C) in Weiss mini-incubation chambers in a recirculating aquaculture system (RAS) in the Wąsosze Fish Farm, near Konin. Dissolved oxygen, water temperature, and pH were maintained at 9.5 mg L^−^^1^, 16 °C, and 8.0, respectively, throughout the experimental period. The mean concentration of ammonium nitrogen and nitrites contents was below 0.01–0.05 mg L^−^^1^ throughout the experimental period. The photoperiod during the experimental trial was maintained at 12L:12D.

### 2.3. Embryo Sampling

The fertilized eggs from three females were incubated in three groups (group A, group B, and group C), and 20 embryos representing nine distinct developmental stages (before larval hatching) were sampled during incubation. Embryos were sampled (nine sampling sessions) from groups A, B, and C (2.0, 5.5, 13.0, 20.0, 24.0, 26.0, 35.0, 55.0, and 160.0 hours postfertilization—hpf). Every sample from each group (groups A, B, and C) was placed in 4% paraformaldehyde in Eppendorf tubes labeled with a letter denoting the female (A, B, or C) and the number of the sampling session (1–9). 

### 2.4. Observations of Embryonic Development and Acquisition of Microscopic Images

The samples were placed on Petri plates and observed under the Olympus SZX16 stereoscopic microscope (Olympus, Tokyo, Japan) equipped with the Olympus DP25 camera (Olympus, Tokyo, Japan). The Olympus KL2500 LCD (Olympus, Tokyo, Japan) fiber optic light source was used for optimal sample imaging. The samples were observed, and images were acquired with the use of Olympus CellSens Dimension software (Olympus, Tokyo, Japan).

### 2.5. Identification of Embryonic Development Stages

The stages of embryonic development were identified by analyzing the structures characteristic of each stage. A list of 36 distinct stages of embryonic development in Russian sturgeon (*Acipenser gueldenstaedtii*) compiled by Dettlaff and Vassetzky [9] was used for this purpose, and the number of developmental stages was reduced from 36 to 30 for Siberian sturgeon [10]. The survival rates of embryos in all analyzed developmental stages were also determined. 

### 2.6. Statistical Analysis

The results of survival are presented as the mean ± SD, and the differences were considered significant at *p* < 0.05. Percentages data were transformed using arc-sin transformation prior to statistical analysis. Before analyses, normality and lognormal tests were also performed, and the normal distribution of samples was checked. Ordinary two-way ANOVA with main effects (Šídák’s multiple comparisons test) was provided.

## 3. Results

Images of each embryonic development stage were acquired for analysis (Figure 1, Figure 2 and Figure 3). No differences were observed between the developmental stages of Siberian sturgeon embryos and the embryos of Siberian sturgeon x Russian sturgeon hybrids. However, minor differences in the timing of the analyzed stages were observed between all females (Figure 1, Figure 2 and Figure 3). The evaluated stages of embryonic development are compared in Figure 4. Embryo survival rates were determined in each developmental stage. The survival rate of hybrid embryos was highest in group B and lowest in group A, whereas Siberian sturgeon embryos were characterized by moderate survival rates relative to groups A and B (Figure 5). During gastrulation, embryo survival rates reached 86.7% in group A, 93.8% in group B, and 91.7% in group C.

## 4. Discussion

Sturgeon hybrids are characterized by high values of performance, breeding, and reproductive parameters, and their rearing is often more economically efficient than the rearing of purebred species [6,11]. The high-productive performance of sturgeon hybrids can be directly attributed to desirable traits such as a high growth rate, early sexual maturation, higher larval survival rates, and improved meat taste and consistency. The most successful sturgeon hybrids are obtained by crossing species with the same number of chromosomes, such as Siberian sturgeon x Russian sturgeon, and vice versa. Sturgeon hybridization programs have been conducted in Poland [12,13], and the resulting hybrids were characterized by superior growth parameters relative to the parent species [13]. 

Controlled breeding and egg incubation are key stages during breeding and hybridization programs. During incubation, embryos proceed through a series of developmental stages, and larvae are hatched. Embryogenesis and successive developmental stages should be closely monitored to identify any irregularities. In purebred sturgeons and hybrids, the main stages of embryonic development are highly similar, but sturgeon species have different thermal requirements, which can affect the rate of embryogenesis. Therefore, embryonic development in purebred sturgeon and hybrids should be thoroughly investigated to monitor all stages of the process and identify any abnormalities. 

In comparison with teleosts, sturgeon species differ in the main stages of the life cycle, in particular embryonic development [7,10]. Sturgeon embryogenesis exhibits an uneven pattern of holoblastic cleavage. This is characterized by the vegetal hemisphere that is not completely divided with each cleavage furrow [10]. 

Developmental defects can be observed in different stages of embryogenesis. According to Dettlaff et al., [14], most developmental defects in sturgeon embryos result from disruptions in morphogenetic movement during gastrulation [15,16,17]. If the epiboly at the vegetal pole is inhibited prematurely during gastrulation, the embryo will die. If the yolk plug is not covered during neural plate formation (primordium of the neural tube), embryos may survive until hatching, but they will have considerable morphological defects. Gastrulation defects can also be caused by aberrant cleavage divisions and suboptimal conditions during incubation [15,16,17]. In the present study, developmental defects were not observed in any of the analyzed groups, which testifies to the high quality of gametes (eggs and sperm) and optimal conditions during embryo incubation (water temperature, dissolved oxygen concentration, and light access). 

Most developmental defects result from abnormalities in the earliest stages of embryogenesis (oocyte maturation and fertilization). Research into several sturgeon species has demonstrated that abnormalities in later stages of embryogenesis can be caused by inadequate environmental conditions and poor water quality [10,14,18]. The percentage of normal embryos (relative to dead and defective embryos) is determined mainly by egg and sperm quality, egg survival rate, and environmental conditions. High-quality eggs are homogeneous, and cell divisions occur synchronously. Initially, the survival rate of incubated eggs can be determined under the microscope during the second and third stages of cell division (four to eight blastomeres).

In the current study, Siberian sturgeon embryos and the embryos of Siberian sturgeon × Russian sturgeon hybrids developed normally. The embryonic development of Siberian sturgeon (*Acipenser baerii*) has been previously analyzed by Park et al., [10]. The cited authors examined 30 developmental stages, from fertilization to hatching, which were identified based on the data collected by Dettlaff and Vassetzky [9] for Russian sturgeon (*Acipenser gueldenstaedtii*), with some modifications. Dettlaff and Vassetzky [9] described the stages of embryonic development in Siberian sturgeon in greater detail than Gisbert et al. [19]. They identified 36 stages of embryonic development, analyzed the influence of water temperature on embryogenesis, and described morphological changes in hatched larvae.

The embryo survival rate was higher in group B hybrids than in the control group (Siberian sturgeon), whereas group A hybrids were characterized by lower survival rates than group C embryos. These results suggest that the survival of hybrid embryos could be determined by individual traits, such as gamete (eggs and sperm) quality and genetic factors. Eggs’ quality was assessed as very high based on morphological traits, and the influence of genetic factors could not be ruled out. In view of the above, further research is needed to examine the genetic profile of the parent species [20] selected to produce hybrids and to analyze genetic diversity in hybrid offspring relative to the parent species.

## 5. Conclusions

Currently, the production of sturgeon fish shows an increasing trend. In addition to pure species, interspecies sturgeon hybrids are also bred. The production of sturgeon hybrids is important in aquaculture because these hybrids very often show favorable breeding traits that are desired by fish farmers. This paper describes the embryonic development of Siberian sturgeon × Russian sturgeon, which is the most common interspecies hybrid used in European aquaculture. During the trial, the most important developmental stages were described, and based on these descriptions it is possible to confirm the accuracy of the embryogenesis of the studied hybrids.

## Figures and Tables

**Figure 1 animals-13-00042-f001:**
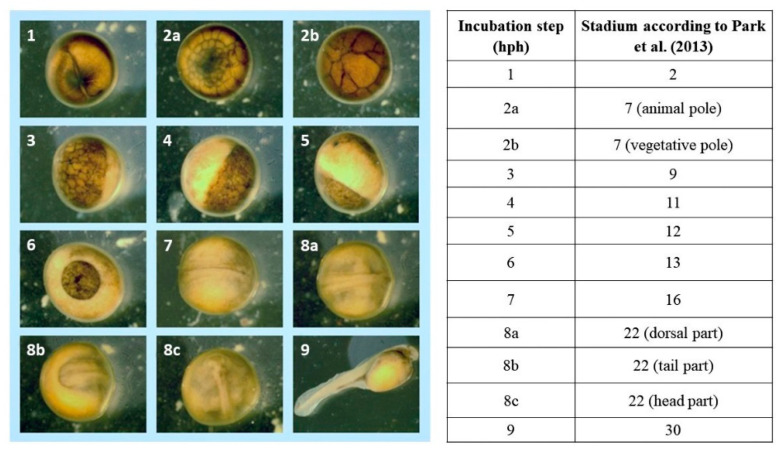
Embryonic development of hybrids [♀ Siberian sturgeon × Russian sturgeon] × ♂ Siberian sturgeon offspring (group A) according to Park et al. [10].

**Figure 2 animals-13-00042-f002:**
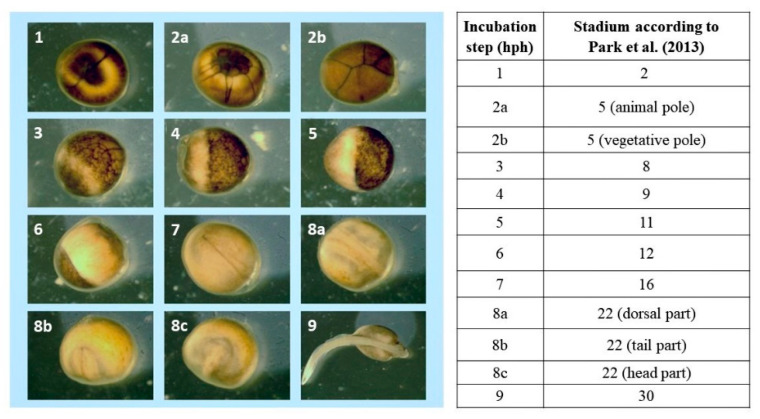
Embryonic development of hybrids [♀ Siberian sturgeon × Russian sturgeon] × ♂ Siberian sturgeon offspring (group B) according to Park et al. [10].

**Figure 3 animals-13-00042-f003:**
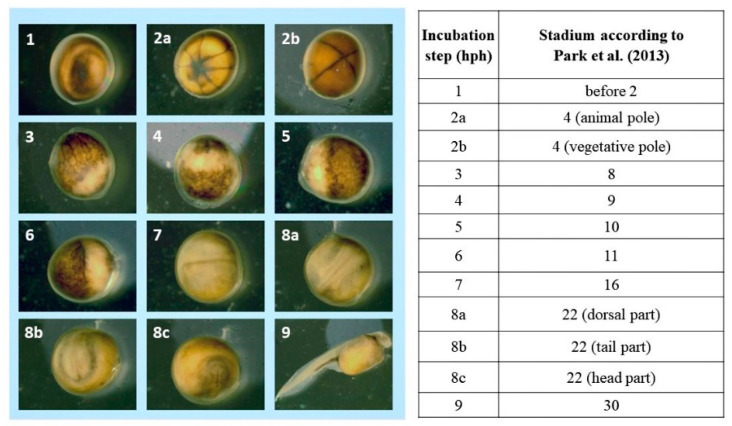
Embryonic development of Siberian sturgeon (♀ Siberian sturgeon × ♂ Siberian) offspring (group C—control) according to Park et al. [10].

**Figure 4 animals-13-00042-f004:**
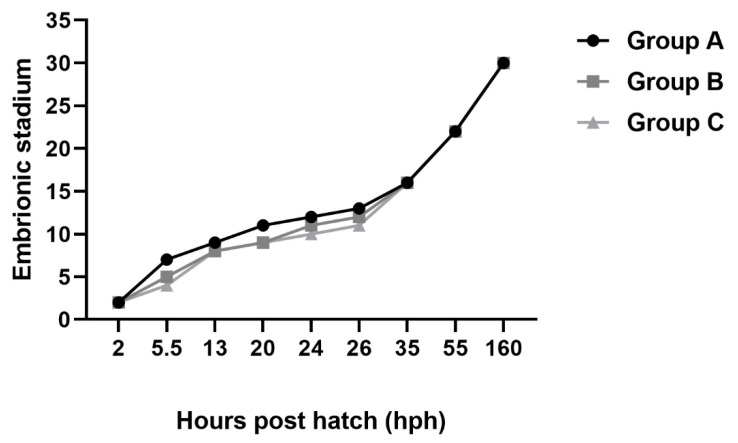
Comparison of embryonic development between hybrid sturgeons in experimental groups.

**Figure 5 animals-13-00042-f005:**
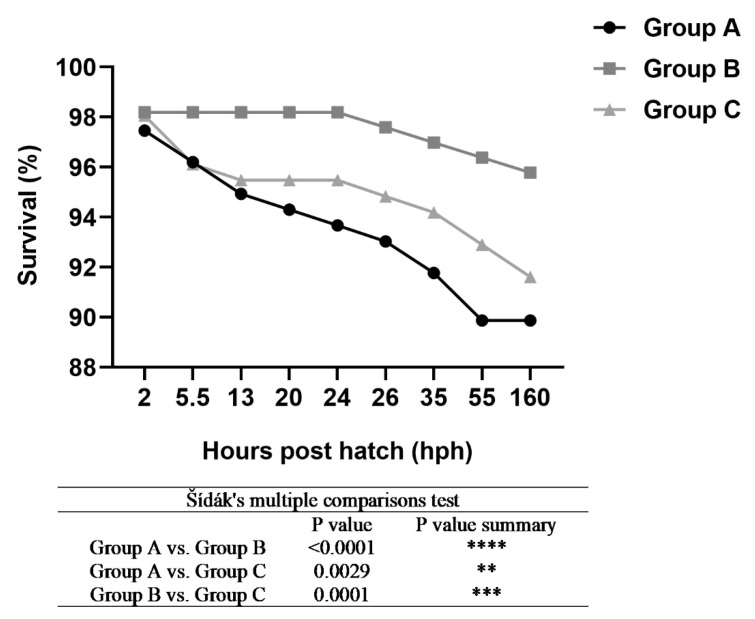
Survival rate of hybrid sturgeons in experimental groups.

## Data Availability

The data presented in this study are available on request from the corresponding author. The data are not publicly available for privacy reasons.

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
