# Peer review of "Embryonic Development and Survival of Siberian Sturgeon × Russian Sturgeon (Acipenser baerii × Acipenser gueldenstaedtii) Hybrids Cultured in a RAS System"

_animals, 2022, doi:10.3390/ani13010042_

Round 1
Reviewer 1 Report
1. In Title: incorrect scientific name
of Siberian sturgeon: bareii instead baerii
2. Fig. 3 - there is pure Siberian sturgeon
But not hybrid as in manuscript.
:
Author Response
Response to Review Comments
(Manuscript ID: animals-2057421)
We are very thankful to the reviewers for their valuable and in-depth review of the manuscript. We have revised our paper as per the reviewer’s suggestions and comments. The author's response to specific comments/suggestions are as follows.
Response to Reviewer #1 Comments
- In Title: incorrect scientific name of Siberian sturgeon: bareii instead baerii
Corrected in the revised manuscript.
- Fig. 3 - there is pure Siberian sturgeon but not hybrid as in manuscript.
Corrected in the revised manuscript.
Reviewer 2 Report
The manuscript, entitled 'Embryonic development and survival of Siberian sturgeon × Russian sturgeon (Acipenser bareii × Acipenser gueldenstaedtii) hybrids cultured in a RAS system', is described in an unclear manner.
The Introduction section describes the context appropriately.
Update the materials and methods with a section on statistical analysis and include the materials used for the experiments.
Author Response
Response to Review Comments
(Manuscript ID: animals-2057421)
We are very thankful to the reviewers for their valuable and in-depth review of the manuscript. We have revised our paper as per the reviewer’s suggestions and comments. The author's response to specific comments/suggestions are as follows.
Response to Reviewer #2 Comments
Update the materials and methods with a section on statistical analysis and include the materials used for the experiments.
Response:
Based on our results we chose ordinary two-way ANOVA with main effects only repeated measures choices are only available when we enter replicate data into subcolumns. Our data contains no replicates and only a main effect model (with no interaction) can be fit. We added this information to MS (Material and Methods) and in Figure 5.
Reviewer 3 Report
I reviewed the article “Embryonic development and survival of Siberian sturgeon × Russian sturgeon (Acipenser bareii × Acipenser gueldenstaedtii) hybrids cultured in a RAS system” from Dorota Fopp-Bayat and colleagues. The paper reports the comparison between different sturgeon hybrids common in the global sturgeon aquaculture. In my opinion, the reported data support the performed trial. I think that the paper is acceptable for the publication after revision.
Therefore, I would like to suggest some improvements before the publication.
General comments:
Lines 13 – 25: Like a simple summary, I suggest to rewrite it using different words of the abstract, they are more or less the same.
Line 40: Please change keywords that are already reported in the title.
Line 44: Please change “in particular” to “mainly”
Line 57: If it is related to the aquaculture production I would like to suggest to change “phenomenon” to “procedures”
Line 78: Please add scientific name of Siberian sturgeons
Lines 85 – 90: Please rephrase it adding the data on the hybrids (lines 89 – 90) in the text
Line 95: Please change “they were” to “than”
Lines 96 – 97: Please change “in a total amount of 5 mg kg-1” to “(5 mg kg-1)”
Line 97: Please change “administered” to “treated”
Line 98: Please change “with” to “by”
Line 99 – 100: Please change “by analyzing the movement of” to “to evaluate the”
Line 112: Please put time point in brackets
Line 114: Why the authors used paraformaldehyde and not buffered formalin? Please explain
Line 115: Please delete “female” from the brackets
Lines 117 – 121: Please add more information on the used equipment (Manufacturer, City, Country)
Line 141: Please delete “3.2. Figures”
Figures: Please provide more clear figures (1-5) and improve the size of that to clear explain the differences
Lines 144 – 145: Please rephrase “Embryonic development of hybrids of ♀ Siberian sturgeon x Russian sturgeon] x ♂ Siberian sturgeon offspring (Group A).” to “Embryonic development of hybrids [♀ Siberian sturgeon x Russian sturgeon] x ♂ Siberian sturgeon offspring (Group A)”, the same in lines 148 - 149.
Lines 157 – 158: Please change “of ” to “between”.
Lines 225 – 230: Please rephrase this paragraph because to many times “sturgeon” is repeated
Line 230: Please change “experiment” to “trial”
Lines 240 – 242: Please report the accession number or reference about the Local Ethical Committee authorization
References: Please check and report the reference list according to the journal requirements.
Author Response
Response to Review Comments
(Manuscript ID: animals-2057421)
We are very thankful to the reviewers for their valuable and in-depth review of the manuscript. We have revised our paper as per the reviewer’s suggestions and comments. The author's response to specific comments/suggestions are as follows.
Response to Reviewer #3 Comments
I reviewed the article “Embryonic development and survival of Siberian sturgeon × Russian sturgeon (Acipenser bareii × Acipenser gueldenstaedtii) hybrids cultured in a RAS system” from Dorota Fopp-Bayat and colleagues. The paper reports the comparison between different sturgeon hybrids common in the global sturgeon aquaculture. In my opinion, the reported data support the performed trial. I think that the paper is acceptable for the publication after revision.
Therefore, I would like to suggest some improvements before the publication.
General comments:
Lines 13 – 25: Like a simple summary, I suggest to rewrite it using different words of the abstract, they are more or less the same.
Response: Simple summary was rewritted during suggestions. Line 40: Please change keywords that are already reported in the title.
Response: Corrected in the revised manuscript
Line 44: Please change “in particular” to “mainly”
Response: Corrected in the revised manuscript
Line 57: If it is related to the aquaculture production I would like to suggest to change “phenomenon” to “procedures”
Response: Corrected in the revised manuscript
Line 78: Please add scientific name of Siberian sturgeons
Response: Corrected in the revised manuscript
Lines 85 – 90: Please rephrase it adding the data on the hybrids (lines 89 – 90) in the text
Response: Corrected in the revised manuscript
Line 95: Please change “they were” to “than”
Response: Corrected in the revised manuscript
Lines 96 – 97: Please change “in a total amount of 5 mg kg-1” to “(5 mg kg-1)”
Response: Corrected in the revised manuscript
Line 97: Please change “administered” to “treated”
Response: Corrected in the revised manuscript
Line 98: Please change “with” to “by”
Response: Corrected in the revised manuscript
Line 99 – 100: Please change “by analyzing the movement of” to “to evaluate the”
Response: Corrected in the revised manuscript
Line 112: Please put time point in brackets
Response: Corrected in the revised manuscript
Line 114: Why the authors used paraformaldehyde and not buffered formalin? Please explain
Response:
Usually, we use 4% paraformaldehyde for embryo fixation. We fix it according to our laboratory protocol that has been used for many years. But, 4% paraformaldehyde can substitute 10% formalin (it's almost the same).
Line 115: Please delete “female” from the brackets
Response: Corrected in the revised manuscript
Lines 117 – 121: Please add more information on the used equipment (Manufacturer, City, Country)
Response: Corrected in the revised manuscript
Line 141: Please delete “3.2. Figures”
Response: Corrected in the revised manuscript
Figures: Please provide more clear figures (1-5) and improve the size of that to clear explain the differences
Response: Corrected in the revised manuscript
Lines 144 – 145: Please rephrase “Embryonic development of hybrids of ♀ Siberian sturgeon x Russian sturgeon] x ♂ Siberian sturgeon offspring (Group A).” to “Embryonic development of hybrids [♀ Siberian sturgeon x Russian sturgeon] x ♂ Siberian sturgeon offspring (Group A)”, the same in lines 148 - 149.
Response: Corrected in the revised manuscript
Lines 157 – 158: Please change “of ” to “between”.
Response: Corrected in the revised manuscript
Lines 225 – 230: Please rephrase this paragraph because to many times “sturgeon” is repeated
Response: Corrected in the revised manuscript
The text was rephrased according to Reviewers’ suggestions.
Line 230: Please change “experiment” to “trial”
Response: Corrected in the revised manuscript
Lines 240 – 242: Please report the accession number or reference about the Local Ethical Committee authorization
Response: Corrected in the revised manuscript
The numer of Local Ethical Committee authorization was included in the text of manuscript.
Local Ethical Committee authorization 75/2012
References: Please check and report the reference list according to the journal requirements.
Response: Corrected in the revised manuscript